# Prognostic Value of Tumor Size in Resected Stage IIIA-N2 Non-Small-Cell Lung Cancer

**DOI:** 10.3390/jcm9051307

**Published:** 2020-05-01

**Authors:** Chih-Yu Chen, Bing-Ru Wu, Chia-Hung Chen, Wen-Chien Cheng, Wei-Chun Chen, Wei-Chih Liao, Chih-Yi Chen, Te-Chun Hsia, Chih-Yen Tu

**Affiliations:** 1Division of Pulmonary and Critical Care Medicine, Department of Internal Medicine, China Medical University Hospital, Taichung 40447, Taiwan; cychen0808@gmail.com (C.-Y.C.); d18351@mail.cmuh.org.tw (B.-R.W.); d7996@mail.cmuh.org.tw (C.-H.C.); d14321@mail.cmuh.org.tw (W.-C.C.); d8040@mail.cmuh.org.tw (W.-C.C.); d1914@mail.cmuh.org.tw (T.-C.H.); d7855@mail.cmuh.org.tw (C.-Y.T.); 2School of Medicine, China Medical University, Taichung 40402, Taiwan; 3Department of Respiratory Therapy, China Medical University, Taichung 40402, Taiwan; 4Hyperbaric oxygen therapy center, China Medical University Hospital, Taichung 40447, Taiwan; 5Department of Surgery, Chung Shan Medical University Hospital, Taichung 40201, Taiwan; cshy1566@csh.org.tw

**Keywords:** non-small-cell lung cancer, stage IIIA-N2, surgery

## Abstract

The eighth edition of the American Joint Committee on Cancer (AJCC) staging system for lung cancer was introduced in 2017 and included major revisions, especially of stage III. For the subgroup stage IIIA-N2 non-small-cell lung cancer (NSCLC), surgical resection remains controversial due to heterogeneous disease entity. The aim of this study was to evaluate the clinicopathologic features and prognostic factors of patients with completely resected stage IIIA-N2 NSCLC. We retrospectively evaluated 77 consecutive patients with pathologic stage IIIA-N2 NSCLC (AJCC eighth edition) who underwent surgical resection with curative intent in China Medical University Hospital between 2006 and 2014. Survival analysis was conducted, using the Kaplan–Meier method. Prognostic factors predicting overall survival (OS) and disease-free survival (DFS) were analyzed, using log-rank tests and multivariate Cox proportional hazards models. Of the 77 patients with pathologic stage IIIA-N2 NSCLC examined, 35 (45.5%) were diagnosed before surgery and 42 (54.5%) were diagnosed unexpectedly during surgery. The mean age of patients was 59 years, and the mean length of follow-up was 38.1 months. The overall one-, three-, and five-year OS rates were 91.9%, 61.3%, and 33.5%, respectively. Multivariate analysis showed that tumor size <3 cm (hazards ratio (HR): 0.373, *p* = 0.003) and video-assisted thoracoscopic surgery (VATS) approach (HR: 0.383, *p* = 0.014) were significant predictors for improved OS. For patients with surgically treated, pathologic stage IIIA-N2 NSCLC, tumor size <3 cm and the VATS approach seemed to be associated with better prognosis.

## 1. Introduction

Lung cancer is the most commonly diagnosed cancer and the leading cause of cancer death in the world. In 2018, an estimated 2.1 million new cases (1,368,524 in men and 725,352 in women) of lung and bronchial cancer were diagnosed, and 1.8 million individuals (1,184,947 in men and 576,060 in women) were expected to die of the tumor [1]. Despite recent advances in molecularly targeted therapy and immunotherapy, the long-term survival of patients with lung cancer remains poor, and the five-year-survival rate is below 20% [2,3]. While more than 80% of tumors were unresectable, surgical resection is the major treatment modality for curative intent, with the five-year survival rate being about 60% [4].

The most important prognostic factor for lung cancer is the stage at presentation, which also guides the clinical management of these patients. Based on a global database of lung-cancer cases assembled by the International Association for the Study of Lung Cancer (IASLC) [5], the eighth edition of the American Joint Committee on Cancer (AJCC) staging system for lung cancer was published in 2017 [6], and it was implemented in clinical practice worldwide in 2018 [7]. In addition to the reclassification of extra-thoracic disease into M1b and M1c, the most significant change distinguishing the eighth edition from the seventh edition is the modification of T classification, which may result in different stage allocations. In the eighth edition, stages T1–T4 are redefined according to tumor size (T1a  ≤  1  cm; 1 cm < T1b < 2  cm; 2 cm < T1c < 3  cm; 3 cm < T2a < 4  cm; 4 cm < T2b < 5  cm; 5 cm < T3 < 7  cm; T4 > 7  cm). For patients with former stage IIIA-N2 disease, the reclassification of tumor size more than 5 cm shifting from T2b to T3 (> 5 cm but < 7 cm) and from T3 to T4 (> 7 cm) results in a change of stage from IIIA to IIIB.

Due to heterogeneous disease entity, the role of surgical resection for patients with former stage IIIA-N2 non-small-cell lung cancer (NSCLC) remains controversial. According to the guidelines [7,8], multidisciplinary team assessment prior to treatment is warranted to evaluate the resectability, depending on single N2 lymph node station involvement and/or small lymph node size (<3 cm). The treatment options include resection, followed by adjuvant chemotherapy; induction therapy, followed by surgery; definitive concurrent chemoradiation; and consolidation therapy with Durvalumab. However, despite the complexity in treatment planning and major changes in T description and stage allocation of the eighth edition, the guidelines do not address the consequent changes to treatment algorithms for patients with clinical stage IIIA-N2 NSCLC. Furthermore, the role of surgical resection with curative intent in such patients has not been well evaluated. Hence, the aim of this study was to evaluate the clinical features and surgical–pathological factors that affect the prognosis of patients with resected stage IIIA-N2 NSCLC.

## 2. Materials and Methods

This study was approved by the Institutional Review Board of China Medical University Hospital (CMUH109-REC1-037, date of approval: 11 March 2020), and informed consent was waived because of the retrospective nature of the study.

### 2.1. Inclusion Criteria

From 1 January 2006 to 31 December 2014, 748 patients with lung cancer underwent surgical resection with mediastinal lymph node dissection or sampling at China Medical University Hospital. The tumor-node-metastasis (TNM) staging system was reclassified according to the eighth edition of the AJCC staging system. A total of 77 (10.3%) patients with stage IIIA-N2 NSCLC who underwent surgical resection with curative intent were enrolled in the study. Smoking status was classified as ever (including current and former smoker) or never smoker. Family history of cancer was defined as any first-degree relative diagnosed with any form of cancer. The preoperative staging workup included complete blood count, serum biochemistry, carcinoembryonic antigen (CEA), chest radiography, chest computed tomography (CT) scan, bronchoscopy, and nuclear medicine exam. Patients with positive surgical tumor margin and incomplete medical record were excluded. There were weekly multidisciplinary lung cancer meetings where thoracic radiologists, radiation oncologist, surgeons, and pulmonologists from the China Medical University Hospital jointly reviewed and discussed the management plan of patients with lung cancer.

### 2.2. Surgical Technique

Only patients having the Eastern Cooperative Oncology Group (ECOG) performance of 0 or 1 were considered as surgical candidates, and all surgery was performed with curative intent. All patients underwent surgery either with preoperatively clinical N2 disease or unexpectedly during surgery. Tumor location was analyzed as dichotomous variables (lower versus upper or middle lobes; peripheral (outer one-third of lung field) versus central (inner two-thirds of lung fields)). Induction therapy was defined as preoperative chemotherapy and/or radiotherapy. Adjuvant therapy was defined as treatment with either chemotherapy, radiotherapy, or a combination of both after surgical resection. The type of surgery included standard (pneumonectomy, bilobectomy, or lobectomy) and limited resection (wedge resection or segmentectomy). Mediastinal lymph node dissection or sampling with a minimum of three different stations was performed according to the surgeon’s experience, and all resected lymph nodes were labeled separately. All pulmonary resections were performed either through open thoracotomy or video-assisted thoracoscopic surgery (VATS).

### 2.3. Histopathological Evaluation

All surgical specimens were evaluated for pathologic staging. Histological typing was performed according to the World Health Organization classification. The recorded variables included tumor size, differentiation grade, visceral pleural involvement, lymphovascular permeation, perineural invasion, multiple N2 station, and N2 ratio. Multiple N2 station was defined as lymph node metastasis involving more than one N2 station. N2 ratio was calculated by dividing the total number of metastatic by the total number of N2 lymph nodes examined.

### 2.4. Statistical Analysis

All statistical analyses were performed by using MedCalc Statistical Software version 19.0.7 (MedCalc Software bvba, Ostend, Belgium; https://www.medcalc.org; 2019). Normally and non-normally distributed continuous data were expressed as mean (standard deviation (SD)) and median (interquartile range (IQR)), respectively. Categorical variables were reported as number (%). Overall survival (OS) was defined as the time from the date of pathological diagnosis until the date of death or last follow-up. Disease-free survival (DFS) was defined as the time from the date of pathological diagnosis until the date of recurrence, death, or last follow-up. Survival curves were estimated by using the Kaplan–Meier method. The prognostic factor analyses were performed by log-rank tests and Cox proportional-hazards regression model. Statistical analysis was considered to be significant when the *p*-value was < 0.05.

## 3. Results

### 3.1. Demographic and Clinicopathologic Characteristics

Of the 77 patients with pathologic stage IIIA-N2 NSCLC, 35 were male, and 42 were female, with a mean age of 59 years (SD, 12.2 years; range, 34 to 82 years). Thirty-five (45.5%) patients were diagnosed as N2 disease before surgery, and 42 (54.5%) were diagnosed unexpectedly during surgery. Forty-one (53.2%) patients underwent VATS, and 36 (46.8%) underwent open thoracotomy. The most common histology was adenocarcinoma (62, 80.5%), followed by squamous cell carcinoma (9, 11.7%). The mean size of tumor was 2.9 cm (SD, 1.0 cm). Forty-five (58.4%) patients had tumors of 3 cm or less in diameter, and 32 (41.6%) patients had tumors greater than 3 cm. With respect to lymph node involvement, multiple N2 station was seen in 21 (27.3%) patients and median N2 ratio was 33.3% (IQR, 13.8–50%). Sixty-five (84.4%) patients received adjuvant chemotherapy, of which 23 patients received postoperative radiotherapy. Demographic and clinicopathologic characteristics of the patients are shown in Table 1.

### 3.2. Overall Survival

Figure 1 depicts that the one-, three-, and five-year OS rates were 91.9%, 61.3%, and 33.5%, respectively. The mean length of follow-up was 38.1 months.

In univariate analysis, the median OS was significantly influenced by tumor size. The median OS was 52.0 months (95% CI: 45.3–66.1) in patients with tumors of 3 cm or less, worsening to 32.6 months (95% CI: 23.2–43.6) in patients with tumors greater than 3 cm (log-rank *p* = 0.002) and corresponding to a five-year OS rate of 43.3% and 21.7%, respectively (Figure 2). Moreover, patients with VATS approach had significantly better OS compared with those who received open thoracotomy (five-year OS: 63.5% vs. 18.3%; log-rank *p* = 0.009). On the other hand, OS rates were significantly worse in patients with elder age (versus those with age under 65 years, five-year OS: 24.2% vs. 39.0%; log-rank *p* = 0.031) and those with ECOG 1 (versus those with ECOG 0, 5-year OS: 19.3% vs. 49.4%; log-rank *p* = 0.016).

Multivariate analysis showed that tumor size <3 cm (HR: 0.373, 95% CI: 0.194–0.714, *p* = 0.003) and VATS approach (HR: 0.383, 95% CI: 0.178–0.824, *p* = 0.014) were significant predictors for OS. Univariate and multivariate data are shown in Table 2 and Table 3.

### 3.3. Disease-Free Survival

The one-, three-, and five-year DFS rates were 53.4%, 24.5%, and 12.5%, respectively.

In univariate analysis, the median DFS was significantly influenced by tumor size. The median DFS was 18.4 months (95% CI: 11.9–33.6) in patients with tumors of 3 cm or less, worsening to 11.0 months (95% CI: 7.1–15.6) in patients with tumors greater than 3 cm (log-rank *p* = 0.016) and corresponding to a three-year DFS rate of 33.4% and 12.5%, respectively (Appendix A
Appendix A). There was a non-significant trend between poor prognosis and both clinical N2 disease (versus unsuspected N2 disease, three-year DFS: 16.2% vs. 31.1%; log-rank *p* = 0.077) and elevated CEA level (versus CEA level less than 3 ng/mL, three-year DFS: 18.2% vs. 33.3%; log-rank *p* = 0.053).

Multivariate analysis showed that tumor size <3 cm (HR: 0.451, 95% CI: 0.235–0.865, *p* = 0.017) and clinical N2 versus unsuspected N2 disease (HR: 2.525, 95% CI: 1.340–4.757, *p* = 0.004) were significant predictors for DFS. Both univariate and multivariate data are shown in Appendix A
Appendix A.

## 4. Discussion

The AJCC TNM staging system is the global standard for lung cancer staging [8]. Compared with the seventh edition, the eighth edition has been validated in several cohorts [9,10], demonstrating better survival stratification and prognosis prediction. With regard to the major changes in the T classification, former stage IIIA-N2 disease is further separated into stage IIIA and IIIB, based on tumor size, which is suggestive of distinct prognosis between the two subgroups. Sui et al. [9] retrospectively analyzed a Chinese cohort including 3599 patients with pathological stage IA to IIIA between 2005 and 2012. Of 772 former stage IIIA patients, stage migration to IIIB was found in 180 (23.3%) patients, and associated with lower five-year survival rate (26.1% vs. 41.7%, *p* < 0.001). Therefore, we focused on updated stage IIIA-N2 NSCLC, which represents a heterogenic group of patients and complex treatment modalities, including surgical resection.

The role of surgical resection for patients with stage IIIA-N2 NSCLC remains controversial, with different management preferences between Europe and America [11]. In Europe, surgeons tend to perform upfront resection, without induction therapy, for single-station, non-bulky N2 disease. The European Society for Medical Oncology (ESMO) guideline recommends that surgical resection, followed by adjuvant chemotherapy, is a reasonable treatment option for single-station N2 disease [8]. By contrast, in America, the standard treatment has been induction chemotherapy or chemoradiation, followed by surgical resection. A Cardiothoracic Surgery Network survey [12] demonstrated that more than 80% of thoracic surgeon preferred induction therapy for stage IIIA-N2 NSCLC, whereas only 12% preferred surgical resection followed by adjuvant therapy. For macroscopic single station N2 disease, 62% would consider surgical resection only if N2 clearance was achieved, whereas 18% considered this inoperable and offer definitive concurrent chemoradiation. Regarding the preference of induction therapy followed by surgical resection in America, considerations include better tolerance to full-dose chemotherapy preoperatively, better control of the systemic micro-metastases, assessment of treatment response before decision of surgery, and possible parenchymal sparing surgery [13]. Therefore, the approach of induction therapy is supported by the National Comprehensive Cancer Center Network (NCCN) guideline [7]. However, despite the high agreement and guideline recommendation, substantial variation in clinical practice existed in The Society of Thoracic Surgeons General Thoracic Surgery Database [14]. Of 3319 clinical stage III-N2 patients, 54% received direct surgical resection and 46% received induction therapy, with five-year survival rates of 36% and 35%, respectively. Considering the controversial role of surgical resection for patients with updated stage IIIA-N2 NSCLC, our study was aimed to investigate prognostic factors to guide therapeutic decisions.

For former stage IIIA-N2 NSCLC, previous studies have well demonstrated prognostic factors, including number of positive lymph nodes [15], microscopic N2 [16], single-station N2 [17,18,19,20], VATS approach [21], lobectomy approach [22], postoperative radiotherapy [23,24], and pathological response after induction therapy [25,26]. However, there is still some concern about changes of T classification and stage migration in the eighth edition. In the study, we presented a single-center retrospective study of 77 surgically resected IIIA-N2 NSCLC patients, staged according to the eighth edition of the AJCC staging system. Our first finding is that tumor size <3 cm was associated with better prognosis (HR: 0.373, *p* = 0.003). The possible reason is that tumor size was correlated with occult systemic micro-metastases. Yang et al. [27] reported that the proportions of cases with N0M0 status with tumor size <2 cm and >7 cm were 70.79% and 33.33%, respectively. Cho et al. [28] analyzed the data of 1821 patients with clinical N0-1 NSCLC, in which they found that tumor size >3 cm was a common predictor for unsuspected N2 and multiple-station N2 disease. Based on our finding and major changes of T classification, further large-scale studies are warranted to confirm the role of tumor size in patient selection and treatment strategy. Our second finding is that VATS approach was associated with better prognosis (HR: 0.383, *p* = 0.014). Previous studies showed similar results [21,26]. Despite the possible selection bias of our study, the consistency of these findings suggests that the VATS approach can be employed safely, without compromised prognosis.

Our 33.5% five-year OS rate is slightly lower than that of the IASLC database [5], in which the five-year OS rates for clinical and pathological stage IIIA disease are 36% and 41%, respectively. The relatively poorer prognosis in our patients highlights the importance of patient selection and the multimodality treatment approach. First, in patients with stage IIIA-N2 NSCLC undergoing surgical resection, the prognostic value of degree of lymph node involvement has been well documented. The ESMO guideline [8] highlights that single-station N2 disease is the most important features while evaluating resectability. Several studies [17,18,19,20] have also demonstrated that multiple-station N2 involvement indicates a poorer prognosis, regardless of whether the induction therapy is given: five-year OS rate is usually below 25%. Given the poorer prognosis and higher risk of systemic micro-metastases, upfront surgical resection should be avoided in patients with multiple-station N2 disease. However, in our study, 21 (27.3%) patients with multiple-station N2 disease received surgical resection, whereas only 10 (13%) patients received induction therapy. Second, regardless of whether to offer surgical resection, the implementation of multimodality treatment is of most importance [7,8]. There is pooled evidence in a network meta-analysis [29] where patients with stage IIIA-N2 NSCLC treated with single modality treatment of either surgery or radiotherapy alone seemed to have the worst outcomes. Nevertheless, in our study, 12 (15.6%) patients received surgical resection, only without adjuvant therapy. The lack of multimodality treatment would also explain the poorer outcome of the study.

Our study has some limitations. First, given the nature of retrospective analysis, patients in our study were highly selected by multidisciplinary team screening and not representative of all patients with stage IIIA-N2 NSCLC. In addition, it is not possible to answer the question whether upfront surgical resection is superior to other multimodality approaches. Second, the number of cases in our study was small. The uneven distribution of clinicopathologic characteristics (e.g., single- or multiple-station N2) and treatment approaches (e.g., induction therapy) complicated the interpretation, and the statistical power could be limited.

## 5. Conclusions

In conclusion, we retrospectively reviewed the clinical and pathological characteristics of patients with completely resected stage IIIA-N2 NSCLC, according to the eighth edition of the AJCC staging system. Tumor size <3 cm was the only independent factor for better OS and DFS. In addition, the VATS approach was also a good prognostic factor regarding OS rate. These findings may be helpful to identify patients with stage IIIA-N2 NSCLC eligible to surgical resection.

## Figures and Tables

**Figure 1 jcm-09-01307-f001:**
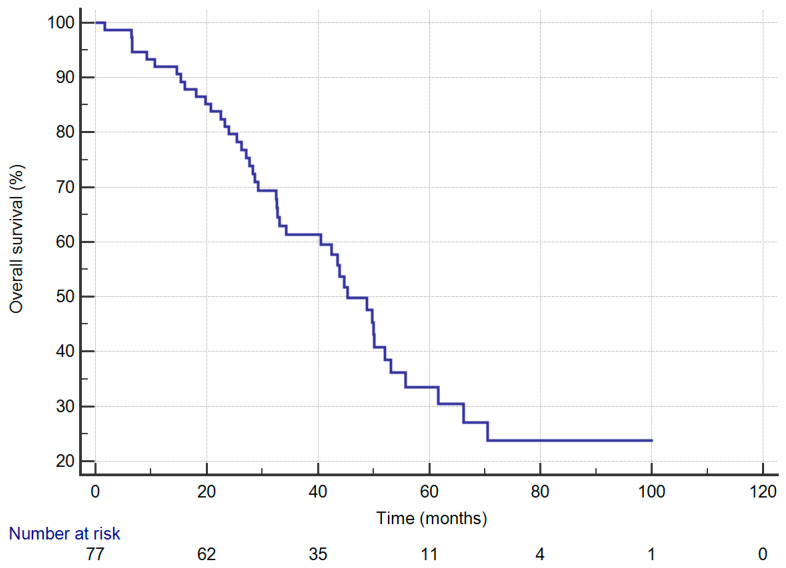
Overall-survival curves of 77 patients with completely resected stage IIIA-N2 non-small-cell lung cancer.

**Figure 2 jcm-09-01307-f002:**
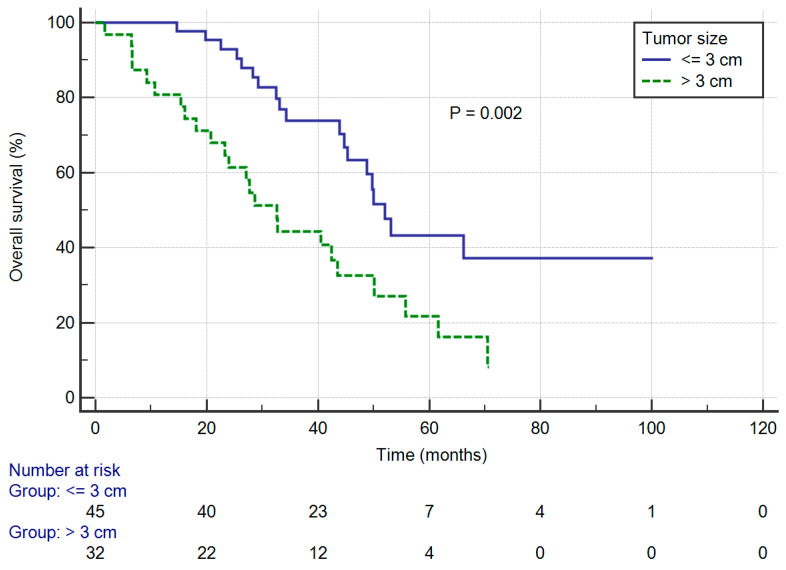
Overall survival curves of patients stratified by tumor size.

**Table 1 jcm-09-01307-t001:** Demographic and clinicopathologic characteristics of 77 patients with resected stage IIIA-N2 non-small-cell lung cancer.

Parameter	Value
Age, Mean (SD), y	59 (12.2)
Male, No. (%)	35 (45.5)
Ever smoker, No. (%)	29 (37.7)
Family History of Cancer, No. (%) Comorbidities, No. (%) Hypertension Cardiovascular Disease Chronic Obstructive Pulmonary DiseaseLiver Disease Chronic Kidney Disease Diabetes Mellitus	14 (18.2) 24 (31.2) 8 (10.4) 6 (7.8) 6 (7.8) 4 (5.2) 10 (13.0)
Performance status, No. (%) ECOG 0 ECOG 1	39 (50.6) 38 (49.4)
Clinical N2, No. (%)	35 (45.5)
Surgical Procedure, No. (%) Limited Resection Standard Resection	8 (10.4) 69 (89.6)
Surgical Approach, No. (%)	
VATS	41 (53.2)
Open Thoracotomy	36 (46.8)
Tumor size, mean (SD), cm	2.9 (1.0)
Tumor Size, No. (%) ≤3 cm 3–5 cm	45 (58.4) 32 (41.6)
Histology, No. (%) Adenocarcinoma Squamous Cell Carcinoma Others	62 (80.5) 9 (11.7) 6 (7.8)
Differentiation, No. (%) Well–moderate Poor Unknown	48 (62.3) 27 (35.1) 2 (2.6)
CEA, Median (IQR), ng/mL	4.0 (2.3–13.1)
Visceral Pleural Involvement, No. (%)	35 (45.5)
Lymphovascular Permeation, No. (%)	66 (85.7)
Perineural Invasion, No. (%) Number of Examined Lymph Nodes, Median (IQR) Number of Positive Lymph Nodes, Median (IQR)	12 (15.6) 14 (9–20) 3 (1–6)
N2 Ratio, Median (IQR), %	33.3 (13.8–50.0)
Tumor Location, No. (%) Central Location Lower Lobe Location	44 (57.1) 31 (40.3)
Multiple N2 Station, No. (%)	21 (27.3)
Induction therapy, No. (%)	10 (13.0)
Adjuvant Therapy, No. (%)	65 (84.4)
Postoperative Radiotherapy, No. (%)	23 (29.9)

SD, standard deviation; IQR, interquartile range; ECOG, Eastern Cooperative Oncology Group; VATS, video-assisted thoracoscopic surgery; CEA, carcinoembryonic antigen; y, years.

**Table 2 jcm-09-01307-t002:** Univariate analysis of factors associated with overall survival.

Parameter	Hazard Ratio (95% CI)	*p*-Value
Age (≥65 y versus <65)	1.939 (1.050–3.582)	0.034
Gender (Male versus Female)	1.084 (0.582–2.020)	0.799
Ever Smoker (Yes versus No)	0.987 (0.526–1.851)	0.967
Family History of Cancer (Yes versus No) Hypertension (Yes versus No) Cardiovascular Disease (Yes versus No) Chronic Obstructive Pulmonary Disease (Yes versus No) Liver Disease (Yes versus No) Chronic Kidney Disease (Yes versus No) Diabetes Mellitus (Yes versus No)	0.681 (0.302–1.532) 0.630 (0.310–1.279) 0.523 (0.161–1.692) 2.094 (0.741–5.916) 1.662 (0.698–3.960) 0.395 (0.054–2.877) 1.040 (0.438–2.469)	0.352 0.201 0.279 0.163 0.251 0.359 0.930
Performance Status (ECOG 1 versus ECOG 0)	2.093 (1.133–3.867)	0.018
Clinical N2 (Yes versus Unsuspected)	0.963 (0.525–1.767)	0.903
Limited Resection (Yes versus Anatomical)	1.453 (0.612–3.449)	0.397
VATS (Yes versus Open Thoracotomy)	0.429 (0.223–0.824)	0.011
Tumor Size (≤3 versus 3–5)	0.390 (0.213–0.715)	0.002
Histology (Adenocarcinoma versus Others)	1.442 (0.689–3.018)	0.332
Differentiation (Poor versus Others)	0.618 (0.311–1.226)	0.168
CEA (≥3 versus <3)	1.593 (0.729–3.482)	0.243
Visceral Pleural Involvement (Yes versus No)	1.359 (0.743–2.486)	0.319
Lymphovascular Permeation (Yes versus No)	1.314 (0.513–3.352)	0.567
Perineural Invasion (Yes versus No)	0.483 (0.173–1.354)	0.166
N2 Ratio (≥40% versus <40%)	1.167 (0.632–2.154)	0.622
Central Location (Yes versus Peripheral)	1.061 (0.576–1.955)	0.848
Lower Lobe Location (Yes versus Upper or Middle)	1.408 (0.757–2.619)	0.280
Multiple N2 Station (Yes versus No)	1.056 (0.550–2.028)	0.870
Induction Therapy (Yes versus No)	0.793 (0.281–2.236)	0.660
Adjuvant Therapy (Yes versus No)	1.147 (0.483–2.725)	0.756
Postoperative Radiotherapy (Yes versus No)	0.551 (0.263–1.151)	0.113

Variables with *p*-values of less than 0.2 were tested in multivariate analysis. CI, confidence interval; ECOG, Eastern Cooperative Oncology Group; VATS, video-assisted thoracoscopic surgery; CEA, carcinoembryonic antigen.

**Table 3 jcm-09-01307-t003:** Multivariate analysis of factors associated with overall survival.

Parameter	Hazard Ratio (95% CI)	*p*-Value
Age (≥65 y versus <65)	1.576 (0.799–3.111)	0.190
Performance Status (ECOG 1 versus ECOG 0)	1.669 (0.878–3.173)	0.118
VATS (Yes versus Open Thoracotomy)	0.383 (0.178–0.824)	0.014
Tumor Size (≤3 versus 3–5)	0.373 (0.194–0.714)	0.003
Differentiation (Poor versus Others)	0.732 (0.358–1.499)	0.394
Perineural Invasion (Yes versus No)	0.681 (0.229–2.023)	0.489
Postoperative Radiotherapy (Yes versus No)	1.173 (0.501–2.745)	0.713

CI, confidence interval; ECOG, Eastern Cooperative Oncology Group; VATS, video-assisted thoracoscopic surgery.

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
