# Peer review of "Prognostic Value of Tumor Size in Resected Stage IIIA-N2 Non-Small-Cell Lung Cancer"

_jcm, 2020, doi:10.3390/jcm9051307_

Round 1

Reviewer 1 Report

In the manuscript Chen. et al. the authors conduct a retrospective study reviewing clinical and pathological characteristics of patients affected by stage IIIA-N2 NSCLC . All the patients were surgically treated with the intention of achieving complete resection.

According to the authors analysis, among all the characteristics analyzed, tumor size <3 cm was the only independent factor affecting both overall survival and disease free survival. They also observe that patients treated though video-assisted thoracoscopic surgery (VATS) have better OS compared to the one treated though open thoracotomy (as already reported).

The analysis performed by the authors has merit, although there are clear limitations that the authors themselves mention in the last part of the discussion. one the main concern is indeed the small number of patients involved in the study. 

Author Response

Reviewer #1 comments:

In the manuscript Chen. et al. the authors conduct a retrospective study reviewing clinical and pathological characteristics of patients affected by stage IIIA-N2 NSCLC . All the patients were surgically treated with the intention of achieving complete resection.

According to the authors analysis, among all the characteristics analyzed, tumor size <3 cm was the only independent factor affecting both overall survival and disease free survival. They also observe that patients treated though video-assisted thoracoscopic surgery (VATS) have better OS compared to the one treated though open thoracotomy (as already reported).

The analysis performed by the authors has merit, although there are clear limitations that the authors themselves mention in the last part of the discussion. One the main concern is indeed the small number of patients involved in the study.

Response: Thanks for your kind comment. The small number of patients is indeed considered as the weakness of our study. Further randomized controlled trial to evaluate the role of surgical resection for patients with stage IIIA-N2 NSCLC is warranted.

Reviewer 2 Report

Dr. Chen and colleagues reported a single-center retrospective analysis of 77 patients with stage IIIA-N2 NSCLC. The authors found that tumor size<3cm and VATS are associated with improved survival.

It is an interesting clinical question. However, the sample size is too small to fit a multivariable model. The risk of type II errors is extremely high. In addition, this study is also associated with the following issues in study design and statistical analysis.

  1. Patients’ comorbidities were not specified and not included in models.
  2. Dichotomize tumor size as <3 vs. 3-5 cm is arbitrary. It’s better to model it as a continuous variable.
  3. The authors found that VATS was a significant predictor of better survival. However, this may be purely due to patient selection. Since VATS is the procedure of choice in treating NSCLC nowadays, surgeons will choose thoracotomy only if the VATS is not feasible, like unusual anatomical locations or technical difficulties during VATS. Not considering these factors makes this finding meaningless.
  4. One important parameter, the number of resected lymph nodes, was not included in the model.
  5. Please provide clear definitions of induction therapy, adjuvant therapy, and postop radiotherapy, as these three are somewhat overlapped with each other.
  6. The authors fitted a Cox proportional hazard regression model. However, the proportional-hazard assumption was not tested properly. The PH assumption needs to be checked for each of the variables in the model. Plotting the log cumulative hazard versus time stratified by the variable of interest could be performed to check the proportional hazard graphically. A more formal test -supremum test could also be used.
  7. Please specify the follow-up information. How many patients were lost to follow-up? The reason for LTFU?

Reviewer 3 Report

Thank you for your manuscript, I've read it carefully and I find it an interesting study. I have some minor questions: 

  • Line 41. Although surgery is the main treatment modality in localized disease, radiotherapy aslo can be curative. 
  • What is the mean/medium number of resected nodes in lymphadenectomy?
  • You don-t mention the the complications related to the surgery, can you provide some information about it?
  • Do you think that the best results with VATs are related to fewer surgical complications, or is it because this procedure has been used more in recent years, with overall improvements in the management of this type of patient?

Thank you for your time. 

Author Response

Reviewer #3 comments:

Thank you for your manuscript, I've read it carefully and I find it an interesting study. I have some minor questions:

  1. Line 41. Although surgery is the main treatment modality in localized disease, radiotherapy also can be curative.

Response 1: Thanks for your kind reminder. We have revised the introduction section as following:

“While more than 80% of tumors were unresectable, surgical resection is the major treatment modality for curative intent, with 5-year-survival rate about 60%.”

  1. What is the mean/medium number of resected nodes in lymphadenectomy?

Response 2: Thank you for the valuable comment. The mean (median) number of lymph nodes resected was 15.4 (14), and the mean (median) number of positive lymph nodes was 4.1 (3).

  1. You don-t mention the complications related to the surgery, can you provide some information about it?

Response 3: Thanks for your kind comment. A total of five complications occurred in four patients (5.2%), including bronchopleural fistula (n = 1), subcutaneous emphysema (n = 1), cerebral infarction (n = 1) and empyema (n = 2).

  1. Do you think that the best results with VATs are related to fewer surgical complications, or is it because this procedure has been used more in recent years, with overall improvements in the management of this type of patient?

Response 4: Thank you for the valuable comment. We agreed with you that VATS as a significant predictor of better survival may result from selection bias. We have revised the discussion section as following:

“Despite the possible selection bias of our study, the consistency of these findings suggests that VATS approach can be employed safely without compromised prognosis.”

Round 2

Reviewer 2 Report

Thank you for addressing my comments.

Please attach the plots for the Schoenfeld residuals for each variable included in the Cox model. 
